# Epidemiology of *Clostridioides difficile* in South Africa

**Pieter de Jager**[1,2]*, **Oliver Smith**[3], **Stefan Bolon**[3], **Juno Thomas**[4], **Guy A. Richards**[5]

1 Department of Anesthesiology, Charlotte Maxeke Johannesburg Academic Hospital, Johannesburg, South Africa, 2 Anesthesiology and Pain Medicine, Mount Sinai Hospital, Toronto, Ontario, Canada, 3 Department of Anaesthesiology, School of Clinical Medicine, University of the Witwatersrand, Johannesburg, South Africa, 4 Head of Centre for Enteric Diseases, National Institute for Communicable Diseases, National Health Laboratory Service, Sandringham, Johannesburg, South Africa, 5 Division of Critical Care, School of Clinical Medicine, University of the Witwatersrand, Johannesburg, South Africa

* ppdejager@gmail.com

## Abstract

### Background

*Clostridioides difficile* (CD) is the most common healthcare-associated enteric infection. There is currently limited epidemiological evidence on CD incidence in South Africa.

### Aim

To estimate the burden of CD infection (CDI) in the South African public sector between 1 July 2016 and 30 June 2017.

### Methods

A retrospective cohort study utilizing secondary data was conducted to describe the epidemiology of CD in South Africa. We assessed the patient-level association between variables of interest, CD, and CD recurrence, by undertaking both univariate and multivariable analysis. Adjusted incidence rate ratios (aIRR) were calculated utilizing multivariable Poisson regression. The incidence of CD, CD recurrence and CD testing was estimated by Poisson regression for various levels of care and provinces.

### Results

A total of 14 023 samples were tested for CD during the study period. After applying exclusion criteria, we were left with a sample of 10 053 of which 1 860 (18.50%) tested CD positive. A positive and significant association between CDI and level of care is found, with patients treated in specialized tuberculosis (TB) hospitals having a five-fold increased adjusted incidence risk ratio (aIRR) for CDI (aIRR 4.96 CI95% 4.08–6.04,) compared to those managed in primary care. Patients receiving care at a secondary, tertiary, or central hospital had 35%, 66% and 41% increased adjusted incidence of CDI compared to those managed in primary care, respectively. National incidence of CDI is estimated at 53.89 cases per 100 000 hospitalizations (CI95% 51.58–56.29), the incidence of recurrence at

**Data Availability Statement:** Data cannot be shared publicly because it contains potentially sensitive information and legal and ethical restrictions apply to sharing the data. Data are available to qualifying organizations and/or

individuals from the National Health Laboratory Service (email: helpdesk1@nhls.ac.za). Demographic and health service data are available from the Health Systems Trust (email: hsr@hst. org.za website: https://www.hst.org.za/contact).

**Funding:** The authors received no specific funding for this work.

**Competing interests:** The authors have declared that no competing interests exist.

21.39 (CI95% 15.06–29.48) cases per 1 000 cases and a recurrence rate of 2.14% (CI95% 1.51–2.94).

## Conclusion

Compared to European countries, we found a comparable incidence of CD. However, our estimates are lower than those for the United States. Compared to high-income countries, this study found a comparatively lower CD recurrence.

## Introduction

*Clostridioides difficile* (CD) is a spore-forming Gram-positive obligate anaerobic bacillus, first identified as part of the commensal gut flora in healthy infants [1]. CD spores are resistant to various environmental factors, including most antimicrobial agents and disinfectants, and can survive on external surfaces for months [2]. Spores are transmitted via the fecal-oral route and are activated in the duodenum by bile salts and L-glycine, acting as a co-germinant CspC receptor on the spore [3]. Germinants induce enzyme-mediated degradation of the spore cortex, releasing the chromosome-containing core and allowing the proliferation of vegetative cells [4]. Once vegetative CD bacilli have proliferated, the mucolytic enzyme Cwp84 degrades mucosal protective barriers in the gut, allowing access to the intestinal epithelium and subsequent colonization of the large intestine [5]. CD expresses several pathogen-associated molecular patterns (PAMPS), including Toxin A (TcdA), Toxin B (TcdB), flagellin, surface layer protein A (SlpA) and peptidoglycan fragments which each stimulate the host innate immune response via pattern recognition receptors, leading to both a local and systemic inflammatory response.

Virulence of CD is primarily mediated by the production of toxins encoded in the pathogenicity locus (PaLoc) [6]. PaLoc codes for three protein toxins: toxin A (TcdA), toxin B (TcdB) and *CD* transferase toxin (CDT) [7]. Of these TcdA and TcdB are more pathogenic, causing receptor-mediated inactivation of Rho with subsequent disruption of the cellular cytoskeleton, breakdown of tight junctions and enterocyte apoptosis [7, 8]. Depending on the virulence of the strain and host immunity, symptoms vary from asymptomatic colonization to fulminant colitis which is associated with high levels of morbidity and mortality [9]. Patients typically present with three or more episodes of watery diarrhea within 24 hours, with or without systemic signs such as fever and dehydration [10]. If the disease progresses to hemorrhagic colitis patients develop bloody diarrhea and severe abdominal pain. Clinical and laboratory markers of severity include body temperature greater than 38.5˚C; severe abdominal pain; an elevated lactate level; elevated serum creatinine level; and a white cell count of greater than $15 \times 10^9$/L [11].

*Clostridioides difficile* infection (CDI) is defined by the Infectious Diseases Society of America (IDSA) and Society for Healthcare Epidemiology of America (SHEA) as 'the presence of symptoms (usually diarrhea) and either a stool test positive for CD toxins or detection of toxigenic *CD*, or colonoscopy or histopathologic findings revealing pseudomembranous colitis' [12].

Notwithstanding the increasing incidence of community-acquired CDI, recent hospitalization and administration of antibiotics remain the main risk factors for acquiring CDI [13]. Additional risk factors include outpatient antibiotic exposure, co-morbidities such as renal failure, malignancies and immunosuppression and age over 65 years [14]. Despite large

variability in the reported incidence of CDI, it is clear that the burden of CDI has grown substantially since the late 1990s, particularly in high-income countries [1]. CDI is associated with significant morbidity and mortality and places an economic burden on healthcare systems [15]. However, our current understanding of the epidemiology of CDI is primarily informed by studies from high-income countries, with very few from South Africa estimating the burden of CDI [16–18]. We, therefore, undertook this study to estimate the incidence of CDI and associated risk factors in the South African public sector.

## Methods

We conducted a retrospective cohort study. Secondary data sources for the period 1 July 2016 to 30 June 2017 were utilized to identify factors associated with CDI and CDI recurrence and to describe the epidemiology in the South African public sector. Data were accessed on 10 July 2020. Findings are reported in line with the Strengthening the Reporting of Observational Studies in Epidemiology (STROBE) Statement.

### Ethics

The protocol for this study was reviewed by the Human Research Ethics Committee (Medical) at the University of the Witwatersrand, Johannesburg and clearance was obtained (Ref: M200114) for this specific study before commencing the study. This was a retrospective study utilizing secondary data sources. Permission was sought and obtained from the NHLS before accessing the data. Before data analysis, all unique identifiers were delinked to ensure anonymity. This study included data on minors over the age of one year.

### Study setting

South Africa is an upper-middle-income country with a 2016 mid-year estimated population of 55.91 million [19]. The health sector has three tiers–a national level, provincial level (9 provinces) and district level (52 districts). Healthcare facilities are divided into community health centers providing primary and community-based care, district hospitals, regional hospitals, tertiary and central hospitals as well as specialized service hospitals (e.g. tuberculosis and psychiatric) [20]. Approximately 84% of South Africans rely on the public healthcare sector to access hospital care with laboratory diagnostic services provided by the National Health Laboratory Services (NHLS) [21, 22].

The NHLS has a national database of all laboratory results of all patients treated in the South African public sector. Routine healthcare utilization and private health insurance coverage rates are collected and annually reported by the District Health Barometer [23].

### Data management and analysis

Data from all patients who had a stool sample submitted to the NHLS between 1 July 2016 and 31 June 2017 for CD testing were considered for inclusion in the study, therefore our sample can be considered representative of the South African public sector. The NHLS only tests unformed (Bristol stool class 5–7) specimens for CD. Testing is conducted utilizing enzyme immunoassay, polymerase chain reaction (PCR), GeneXpert or other PCR-based assays and standard diagnostic protocols. Data on healthcare facility level utilization rates for the period 1 July 2016 to 31 June 2017 were obtained from the District Health Barometer database. To estimate crude national incidence, mid-year population estimates from Statistics South Africa for 2016 were utilized and adjusted for medical insurance coverage.

Data from the NHLS was merged with that from the District Health Barometer to calculate incidence rates. No facility-level public sector utilization data stratified by age and sex, and no utilization data for the private sector were available. To estimate the incidence of CDI in the public sector we excluded tests from the private sector and those from patients under the age of 1 year. There were no missing data in the final data set utilized in the analysis.

For analysis, data were imported into Stata version 16 [24]. Patient ages were recoded into a categorical variable (1–30; 31–40; 41–60; 61 or older). Level of care was defined as primary [district, community health centre or non-infectious disease specialist facility (psychiatric and rehabilitation hospitals)]; secondary (regional hospital, typically providing general medical or surgical specialist-level care); tertiary (providing sub-specialty care), central hospital care (providing quaternary level care) and specialized tuberculosis (TB) hospitals. Finally, a binary variable was coded to compare high dependency care unit (ICU/HC) patients with those receiving care in a general ward (non-ICU/HC).

Categorical variables are reported in terms of frequencies, proportions and percentages, and continuous data in terms of measures of spread and central tendency. Univariate analysis was conducted to assess the relationship between CDI or CDI recurrence and variables of interest using $\chi^2$ or Fishers' Exact test. The level of significance is defined at a two-tailed α of 0.05 for both the univariate and multivariable analyses. In the multivariable analysis, a Poisson regression model was developed to estimate crude and adjusted rate ratios and 95% confidence intervals for factors associated with CDI. Pearson goodness-of-fit testing was done to assess model performance.

The seasonality of CDI over the study period of interest was assessed by constructing an epidemiological curve and the proportion of CDI cases diagnosed for each month (July 2016 – June 2017). All incidence estimates and associated confidence intervals were calculated in Stata version 16 by Poisson regression and a period of one year. National crude incidence was estimated utilizing adjusted 2016 mid-year population estimates and expressed as cases per million population. Provincial and level of care incidence was calculated with healthcare facility utilization data and expressed as cases per 100 000 admissions. The incidence of recurrence is estimated in terms of the CD-positive population and expressed recurrence incidence as cases per 1000 CDI cases.

Finally, a geographic information system (GIS) analysis was undertaken to map the incidence and positivity rates across provinces. GIS analysis was conducted in GeoDa (https://geodacenter.github.io/) and the provincial shapefile was obtained from GDAM (https://gadm.org/index.html). Calculated incidence and positivity rate point estimates were manually entered into GeoDa before dividing these estimates into four quartiles to represent the data graphically.

## Results

Fig 1 summarizes the data management process utilized in assessing factors associated with CDI. A total of 14 023 CD tests were conducted by the NHLS nationally between 1 July 2016 and 30 July 2017. Of these, a total of 2 583 were removed due to duplication, namely a repeat test performed for the same patient within the 14 days, the period that distinguishes therapeutic failure from recurrence. A further 1,388 patients were removed because of age < 1 (n = 1,081) missing data, sex (n = 77); unknown age (n = 4) and tests conducted for private sector patients (n = 226). This resulted in a final sample of 10,053 of which 1,860 (18.50%) and 8,193 (81.5%) were CD positive and negative respectively.

Fig 2 illustrates the epidemiological curve for CDI cases between July 2016 and June 2017. CDI cases varied during the year with the greatest number of cases diagnosed during

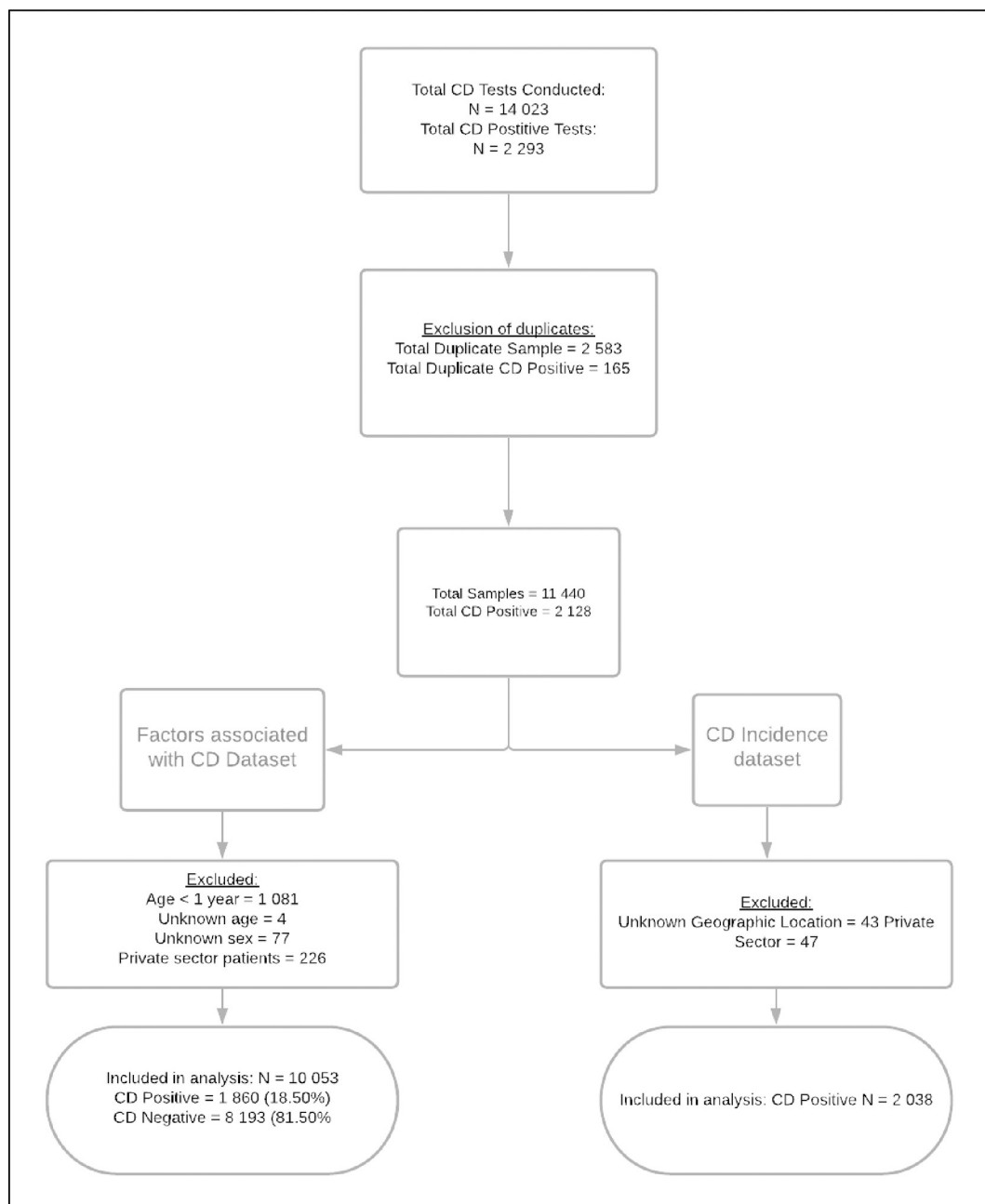

**Fig 1. Summary of data management.**

December (n = 195) and the lowest during May and July (n = 155). From the epidemiological curve, there were slightly more cases during the summer months compared to the winter months.

This analysis found a crude national CDI incidence of 0.101 (CI95% 0.097–0.105) per 1000 000 population and a crude national CD testing incidence of 0.544 (CI95% 0.533–0.554) per 1000 000 population for the South African public sector. Nationally, the estimate of CDI incidence per 100 000 public sector admissions was 53.89 (CI95% 51.58–56.29). Fig 3 summarizes results from the GIS analysis of CD testing, CDI incidence and positivity rates by quintile for

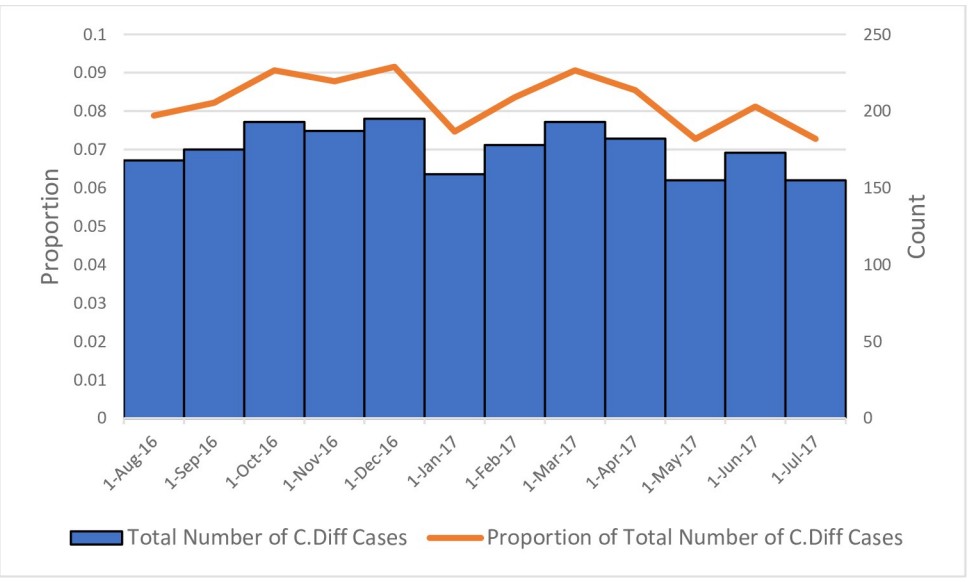

**Fig 2. Seasonal variation in number of CDI cases diagnosed in the South African public healthcare sector 2016/ 2017.** Epidemiological curve for CDI cases between July 2016 and June 2017. CDI cases varied during the year with the greatest number of cases diagnosed during the month of December (n = 195) and the lowest during May and July (n = 155).

various provinces. Incidence varied widely across provinces with Gauteng's incidence of CDI estimated at 158.45 (CI95% 149.69–167.59) per 100 000 admissions followed by the Western Cape at 107.52 (CI95% 98.96–116.62). In contrast, the Northern Cape recorded no CDI cases and for Limpopo province, we estimated an incidence of 0.81 (CI95% 0.26–1.90) cases per 100 000 admissions (Table 1). There was less variation in the positivity rate across provinces. The national average positivity rate was estimated at 18.57% (CI95% 17.85–19.31). The highest positivity rate was recorded in the North West (28.57%) followed by the Eastern Cape (27.84%). The lowest positivity rates were in the Free State (6.65%) and KwaZulu Natal (6.98%).

A higher level of care was associated with a higher incidence of CDI. Central hospitals had a pooled CDI incidence of 230.92 (CI95% 216.43–246.14) per 100 000 admissions, more than double that of tertiary facilities with an estimated incidence of 100.06 (CI95% 91.35–109.37) per 100 000 admissions. A further reduction in the incidence of CDI was found in secondary (20.97 CI95% 18.30–23.92 cases per 100 000 admissions) and primary care (11.85 CI95% 10.23–101.63) facilities.

Central hospitals were also much more likely to test for CD (1 298.46 tests per 100 000 admissions) than tertiary (539.30 tests per 100 000 admissions), secondary 133.45 tests per 100 000 admissions) and primary care facilities (10.87 tests per 100 000 admissions) (Table 2). Therefore, there is less variability in the positivity rate (12.29%– 19.82%) across levels of care than the incidence of CDI. As a group, specialized TB hospitals are a clear outlier with a CDI incidence of 683.98 (CI95%; 589.46–7789.35) cases per 100 000 admissions, a higher testing incidence of 1 115.58 (CI95%; 993.88–1, 248.07) per 100 000 admissions and positivity rates three times greater than the national average (61.31%; CI95% 55.59–66.81).

Of those with CDI, non-ICU/HC, central hospital, female, and patients in the age group 41–60 years had a slightly higher representation (Table 3). The proportion of patients with CDI was significantly higher among those receiving care in a specialized TB hospital (62.01%) compared to tertiary, central, secondary, and primary care facilities respectively at 17.28%, 20.46%, 16.74% and 12.42%.

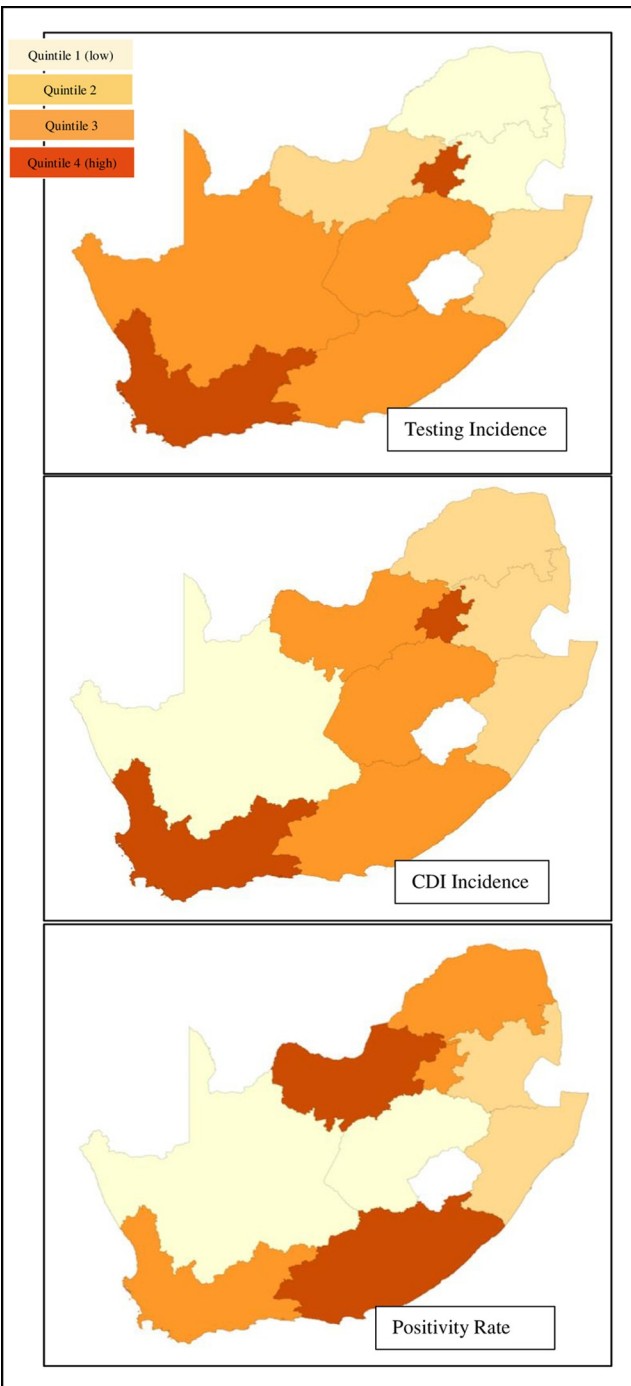

**Fig 3. Provincial clostridioides difficile testing, CDI incidence and positivity rates by quintile: Incidence varied widely across provinces with Gauteng's incidence of CDI estimated at 158.45 per 100 000 admissions followed by the Western Cape at 107.52.** In contrast the Northern Cape recorded no CDI cases and for Limpopo province, we estimated an incidence of 0.81 cases per 100 000 admissions. There was less variation in the positivity rate across provinces. The national average positivity rate was estimated at 18.57%.

Results from the univariate and multivariable analysis are summarized in Table 4. No statistically significant differences were found between age groups, males, and females (IRR 1.03 CI95% 0.94–1.13, p = 0.503) or non-ICU/HC and ICU/HC patients (IRR 0.90 CI95% 0.78–

**Table 1. Incidence of CDI, CD testing and positivity rate by province per 100 000 admissions.**

|  | Total Admission | Total number of CDI Cases | CDI Incidence per 100 000 admissions (CI95%) | Testing Incidence per 100 000 admissions (CI95%) |
|---|---|---|---|---|
| National | 3 781 351 | 2 038 | 37.81 (33–56.29) | 290.29 (284.89–295.78) |
| Eastern Cape | 445 438 | 135 | 30.31 (25.41–35.87) | 108.88 (99.41–119.02) |
| Free State | 207 511 | 22 | 10.60 (6.64–16.05) | 159.51 (142.79–177.65) |
| Gauteng | 770 572 | 1 221 | 158.45 (149.69–167.59) | 801.09 (781.23–821.33) |
| KwaZulu-Natal | 703 882 | 25 | 3.55 (2.30–5.24) | 50.86 (45.72–56.41) |
| Limpopo | 615 364 | 5 | 0.81 (0.26–1.90) | 3.90 (2.50–5.80) |
| North West | 175 116 | 46 | 26.27 (19.23–35.04) | 91.94 (78.29–107.29) |
| Northern Cape | 75 091 | 0 | 0 (-) | 94.55 (73.85–119.26) |
| Western Cape | 541 305 | 582 | 107.52 (98.96–116.62) | 620.91 (600.09–642.26) |
| Mpumalanga | 247 072 | 2 | 0.81 (0.10–2.92) | 5.26 (2.80–9.00) |

1.05, p = 0.186). However, compared to primary care, patients receiving secondary level care were 34% (IRR 1.34 CI95% 1.11–1.64, p = 0.002) more likely to test positive. Similarly, patients receiving care at a tertiary or central hospital were 65% (IRR 1.65 CI95% 1.40–1.94, p < 0.001) and 39% (IRR 1.39 CI95% 1.19–1.62, p < 0.001) more likely to have CDI. Receiving care in a specialized TB hospital was associated with a five-fold increase incidence of CDI (IRR 5.00 CI95% 4.11–6.08, p < 0.001).

After adjusting for age, sex and ICU/HC, the level of care remained statistically significant in its association with CDI with a 35% (aIRR 1.35 CI95% 1.11–1.64, p = 0.003), 66% (aIRR 1.66 CI95% 1.41–1.95, p < 0.001) and 41% (aIRR 1.43 CI95% 1.21–1.65, p < 0.001) increased adjusted incidence compared to primary care for secondary, tertiary, and central hospital care respectively. The strong association with receiving care in a specialized TB hospital also remained statistically significant after adjusting for sex, age, and ICU/HC (aIRR 4.96 CI95% 4.08–6.04, p < 0.001).

After applying the exclusion criteria, 39 cases of CDI recurrence were identified between 1 July 2016 and 30 June 2017. The mean time to recurrence was 43.30 days (CI95% 28.97–57.65) and the probability of a subsequent episode of recurrence was 12.12% (CI95% 3.4–28.2). We estimated the incidence of recurrence to be 21.39 (CI95% 15.06–29.48) cases per 1, 000 cases of CDI. This translates to a recurrence rate of 2.14% (CI95% 1.51–2.94). On univariate analysis, none of the variables under investigation (age, gender, level of care or ICU/HC) were found to be significantly associated with recurrence (Table 5). However, there was a higher incidence of recurrence at specialized TB hospitals (3.11%) and Central hospitals (2.52%) compared to other levels of care (range 1.42% - 1.50%)

**Table 2. Incidence of CDI, CD testing and positivity rate by level of care per 100 000 admissions.**

|  | Total Admissions | Total number of CDI Cases | CDI Incidence per 100 000 admissions (CI95%) | Testing Incidence per 100 000 admissions (CI95%) |
|---|---|---|---|---|
| Central | 408 793 | 944 | 230.92 (216.43–246.14) | 1, 298.46 (1, 263.76–1, 333.88) |
| Tertiary | 484 732 | 485 | 100.06 (91.35–109.37) | 504.82 (485.01–525.22) |
| Secondary | 1 058 310 | 222 | 20.97 (18.30–23.92) | 126.14 (119.47–133.10) |
| Primary | 1 775 613 | 200 | 10.87 (9.39–12.52) | 88.48 (84.15–92.96) |
| Specialized TB | 2 734 | 187 | 683.98 (589.46–789.35) | 1, 115.58 (993.88–1, 248.07) |
| Total | 3 730 182 | 2038 |  |  |

**Table 3. Univariate analysis of factors associated with CDI.**

| | Total (10053) | CDI (1860) | Non-CDI (8193) | p-value |
|---|---|---|---|---|
| | N | n (%) | n (%) | |
| Age (years) | | | | |
| 1–30 | 2, 660 | 477 (25.65) | 2, 183 (26.64) | |
| 31–40 | 2, 427 | 494 (26.56) | 1, 933 (23.59) | |
| 41–60 | 3, 161 | 578 (31.07) | 2, 583 (31.53) | |
| 61 or older | 1, 805 | 311 (16.72) | 1, 494 (18.24) | 0.043 |
| Gender | | | | |
| Female | 5, 693 | 1, 039 (55.86) | 4, 654 (56.80) | |
| Male | 4, 360 | 821 (44.14) | 3, 539 (43.20) | 0.458 |
| Level of Care | | | | |
| Primary | 1, 691 | 210 (11.30) | 1, 481 (18.08) | |
| Specialized TB | 311 | 193 (10.37) | 118 (1.44) | |
| Secondary | 1, 183 | 198 (10.65) | 985 (12.02) | |
| Tertiary | 2, 278 | 466 (25.05) | 1, 812 (22.12) | |
| Central | 4, 590 | 793 (42.63) | 3, 797 (46.34) | <0.001 |
| ICU/HC | | | | |
| Non-ICU/HC | 8, 957 | 1, 675 (90.05) | 7, 282 (88.88) | |
| ICU/HC | 1, 096 | 185 (9.95) | 911 (11.12) | 0.143 |

## Discussion

This is the first study to determine the national incidence of CDI in the South African public sector. CDI incidence is estimated at 53.89 CDI cases per 100 000 hospitalizations, lower than estimates for the US (115.1), but higher than those for England (19.3) [25]. In Europe, the reported incidence of CDI at the healthcare facility level varies between 42 to 1 318 per

**Table 4. Unadjusted and adjusted incidence rate ratios for factors associated with CDI.**

| | Unadjusted IRR (CI95%) | p-value | Adjusted IRR (CI95%) | p-value |
|---|---|---|---|---|
| Age | | | N = 10053 | |
| 1–30 | Reference | | Reference | |
| 31–40 | 1.14 (1.00–1.29) | 0.048 | 1.09 (0.96–1.24) | 0.167 |
| 41–60 | 1.02 (0.90–1.15) | 0.753 | 1.02 (0.90–1.15) | 0.808 |
| 61 or older | 0.96 (0.83–1.11) | 0.584 | 1.02 (0.89–1.18) | 0.741 |
| Gender | | | | |
| Female | Reference | | Reference | |
| Male | 1.03 (0.94–1.13) | 0.503 | 1.00 (0.91–1.09) | 0.974 |
| Level of Care | | | | |
| Primary | Reference | | Reference | |
| Specialized TB | 5.00 (4.11–6.08) | <0.001 | 4.96 (4.08–6.04) | <0.001 |
| Secondary | 1.34 (1.11–1.64) | 0.002 | 1.35 (1.11–1.64) | 0.003 |
| Tertiary | 1.65 (1.40–1.94) | <0.001 | 1.66 (1.41–1.95) | <0.001 |
| Central | 1.39 (1.19–1.62) | <0.001 | 1.41 (1.21–1.65) | <0.001 |
| ICU/HC | | | | |
| Non-ICU/HC | Reference | | Reference | |
| ICU/HC | 0.90 (0.78–1.05) | 0.186 | 0.94 (0.80–1.10) | 0.426 |

**Table 5. Univariate analysis of factors associated with CDI recurrence.**

| | N | Recurrence | No recurrence | p-value |
|---|---|---|---|---|
| | N = 1860 | N = 39 (%) | N = 1821 (%) | |
| Age (years) | | | | |
| 1–30 | 477 | 10 (25.64) | 467 (25.65) | |
| 31–40 | 494 | 13 (33.33) | 481 (26.41) | |
| 41–60 | 578 | 12 (30.77) | 566 (31.08) | |
| 61 or older | 311 | 4 (10.26) | 307 (16.86) | 0.662 |
| Gender | | | | |
| Female | 1, 039 | 21 (53.85) | 1, 018 (55.90) | |
| Male | 821 | 18 (46.15) | 803 (44.10) | 0.871 |
| Level of Care | | | | |
| Primary | 210 | 3 (7.69) | 203 (11.15) | |
| Specialized TB | 193 | 6 (15.38) | 185 (10.16) | |
| Secondary | 198 | 3 (7.69) | 185 (10.16) | |
| Tertiary | 466 | 7 (17.95) | 430 (23.61) | |
| Central | 793 | 20 (51.28) | 687 (37.73) | 0.560 |
| ICU/HC | | | | |
| Non-ICU/HC | 1, 675 | 35 (89.74) | 1, 640 (90.10) | |
| ICU/HC | 185 | 4 (10.26) | 181 (9.90) | 0.792 |

100 000 admissions [26], placing estimates for South Africa at the lower end of the European spectrum.

Previous studies to estimate the incidence of CDI in South Africa are limited. Compared to the central hospitals' incidence in our study of 230.92 per 100 000 admissions, a single centre prospective study in a Cape Town central hospital estimated the incidence of hospital-acquired CDI at 87 per 100 000 admissions in 2013 [17]. A second study from two tuberculosis hospitals in Cape Town found the incidence to be 7007.0 per 100 000 admissions from 2014 to 2015 [18]. This is much higher than the pooled estimate for specialized TB hospitals nationally (683.98 per 100 000 admissions). A possible reason for these differences is that this study's estimates included all public care facilities in South Africa, some of which had no cases of CDI. A facility-level analysis of our data also revealed significant variability in CDI incidence among central hospitals (range 5.35 to 529.36 cases per 100 000 admissions) and specialized TB hospitals (range 0 to 10 661.76 cases per 100 000 admissions). A second possible explanation is that we were unable to differentiate between community and hospital-acquired cases, therefore our estimates of CDI may include cases of community-acquired CDI which may overestimate the incidence. However, a study in 2019 found the prevalence of CD colonization among those living in residential care facilities in Cape Town to be 1.6% [27]. In contrast, an earlier study from a central hospital in Cape Town found that of those patients diagnosed with CDI (N = 59), 32% (n = 19) had community-acquired CDI [17].

There is very limited data on recurrence rates of CDI in South Africa. From this analysis, recurrence is estimated at 2.14% which is lower than that reported in one South African prospective study where three (5.08%) of 59 patients had a recurrence [17]. A large British single-centre prospective study of 2 043 CDI patients with a median follow-up of 11 months found a recurrence rate of 22%, in a quarter of whom the recurrence was with a different CD strain [28]. In addition to advanced age, co-morbidities, healthcare and antibiotic exposure; initial infection with epidemic BI/NAP1/027 CD strain is a risk factor for relapse (as opposed to re-infection) [29]. The prevalence of BI/NAP1/027 was estimated to be 22% between 2013 and 2016 in the

US, and 41.3% in the UK between 2007 and 2008 [30, 31]. There is very sparse data on the epidemiology of BI/NAP1/027 in South Africa. One study found the prevalence to be 3.4% (3/59) [17]. The possible reasons for our apparent low recurrence rates are multiple. Firstly, we only had data for one year which may have excluded possible cases of recurrence at the beginning and the end of the study period. Secondly, some may have been lost to follow-up or died before being diagnosed with a recurrence. Thirdly, South Africa's demographic and population disease profile, antibiotic prescription practices and a lower prevalence of BI/NAP1/027 may have resulted in a true lower incidence of recurrence. Finally, CDI management protocols in South Africa differ from those in countries with higher recurrence rates. Recommended first-line treatment for CDI in South Africa in 2016/17 was oral metronidazole [32] and this was the most common first-line treatment prescribed [33]. Current IDSA recommendations for first-line treatment is either oral vancomycin or fidaxomicin and only where vancomycin or fidaxomicin is not available should metronidazole be considered as a treatment for mild CDI [12]. For recurrences vancomycin or fidaxomicin is administered for longer tapered periods or as pulsed regimens are recommended, and fecal microbiota transplantation can also be considered [12].

An assessment of CDI seasonality suggested a slight increase in cases during the summer months, however, any conclusions regarding this are limited by the fact that we only reviewed data over one year. We are not aware of any other studies from South Africa which have attempted to assess the seasonality of CDI. In contrast to our findings, a systematic review of twenty studies, of which two were conducted in the southern hemisphere, found consistent evidence of seasonality in the incidence of CDI with cases peaking in spring and reducing during summer [34].

Assessment of factors associated with CDI is limited by a lack of potential explanatory variables in this study: age, sex, level of care and high dependency care unit versus ward care. Notwithstanding these limitations, a strong positive association was found between CDI and treatment in a specialized TB hospital, a finding which is support by a previous facility-level analysis in the Western Cape [18]. This association, as well as the association found between CDI and tertiary or quaternary care, can be explained by the fact that patients in these facilities typically suffer from multiple comorbidities and tend to have longer hospital stays and healthcare exposures than primary care patients. Particularly in the case of specialized TB hospitals, patients are more likely to be on prolonged courses of antimicrobial therapy resulting in a disruption of the gut microbiome and subsequent CD proliferation.

## Conclusion

This is the first study in South Africa to estimate the national burden of CDI. Compared to European countries, this analysis found a comparable incidence of CDI. However, estimates are lower than those for the United States. A significantly lower CDI recurrence was found in the South African public sector, compared to high-income countries. Further work is necessary to obtain a clearer picture of the burden of CDI in South Africa. Firstly, the analysis should be extended to include other years and if possible, the private sector. This will allow for the assessment of CDI trends, allow for a more robust analysis of CDI seasonality and a comparison between the public and private healthcare sectors. Secondly, the analysis should be extended to estimate incidence on a district level. Thirdly, the prevalence and trends in the incidence of BI/NAP1/027 should be determined. Of the 14 023 CD tests conducted by the NHLS between 1 July 2016 and 30 June 2017 8, 442 (60.20% utilized GeneXpert technology which routinely reported the presence or absence of BI/NAP1/027 strain. Finally, an economic analysis utilizing existing epidemiological and cost data should be undertaken to estimate the economic burden of CDI on the public healthcare system.

## Supporting information

**S1 Checklist. STROBE statement—checklist of items that should be included in reports of observational studies.**
(DOCX)

## Acknowledgments

National Health Laboratory Service and the District Health Barometer for allowing us access to their data.

## Author Contributions

**Conceptualization:** Pieter de Jager, Oliver Smith, Guy A. Richards.

**Data curation:** Pieter de Jager, Oliver Smith, Juno Thomas.

**Formal analysis:** Pieter de Jager, Oliver Smith.

**Investigation:** Pieter de Jager, Oliver Smith, Guy A. Richards.

**Methodology:** Pieter de Jager, Oliver Smith, Guy A. Richards.

**Project administration:** Pieter de Jager, Oliver Smith, Stefan Bolon, Guy A. Richards.

**Resources:** Pieter de Jager, Oliver Smith, Stefan Bolon, Guy A. Richards.

**Software:** Pieter de Jager.

**Supervision:** Oliver Smith, Stefan Bolon, Guy A. Richards.

**Validation:** Pieter de Jager, Oliver Smith, Stefan Bolon, Guy A. Richards.

**Visualization:** Pieter de Jager, Oliver Smith, Stefan Bolon, Guy A. Richards.

**Writing – original draft:** Pieter de Jager, Oliver Smith, Stefan Bolon.

**Writing – review & editing:** Pieter de Jager, Oliver Smith, Stefan Bolon, Juno Thomas, Guy A. Richards.

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
