## [Decision Letter · Decision Letter 0]

23 Mar 2021

PONE-D-21-03916

Epidemiology of Clostridioides difficile in South Africa

PLOS ONE

Dear Dr. de Jager,

Thank you for submitting your manuscript to PLOS ONE. After careful consideration, we feel that it has merit but does not fully meet PLOS ONE’s publication criteria as it currently stands. Therefore, we invite you to submit a revised version of the manuscript that addresses the points raised during the review process.

We look forward to receiving your revised manuscript.

Kind regards,

Orvalho Augusto, MD, MPH

Academic Editor

PLOS ONE

Additional Editor Comments:

Epidemiology of Clostridioides difficille in South Africa

PONE-D-21-03916

This work is about the incidence of clostridioides difficile infection (CDI) in South Africa (SA). Current knowledge, mostly derived from high income countries, suggests that clostridioides difficile (CD) is a major cause of healthcare associated diarrhea and it is increasingly present in the community. This manuscript contributes to raise clinical awareness of its existence in SA and, I dare say, to the all African continent.

To estimate the incidence, as numerator, the authors took the [newly?] number of cases diagnosed through the National Health Laboratory Service (NHLS) between 1 July 2016 and 30 June 2017. According to the authors the NHLS has a national database of all processed laboratory results in the South Africa public sector [the public health sector which covers 84% of South Africa population]. For the denominator they took the overall mid-year population of South Africa in 2016. Therefore, the CDI incidence is underestimated for both underestimated numerator and large denominator, but I agree with them that it is useful estimate.

In addition to the incidence estimation, the authors produce some measure of association using as exposures (the call risk factors) age, gender, level of care and ICU/HC, and outcomes the state of positive or negative to diagnosis of CDI. Although the negatives here are not good controls it is OK to this exercise.

Few issues:

In the next version please add line numbering. That helps to track and indicate the place of an issue.

I. Introduction is adequate and quite instructive.

II. Methods:

Why the study of risk factors ignore incidence rates? The authors proceed as if they built a case-control with OR estimation. I guess you can obtain population per age and gender. That could be used on a Poisson regression which gives IRR/RR. Just a clarification.

1. Somewhere in the first paragraph there is an indication that a STROBE form was filled. I did not find the attached form. Please add this in the supplements.

2. Data management and analysis

- For the mid-year population conceptually it represents the number of people living in a specific geographic area on 1st July (others would say 30th of June). That is too simplistic. It is more useful to think of it as the total of the fractions year that each individual in South Africa lived. Eg: some lived the full year so their fraction is 1; others just the first quarter so 0.25; and others up to 3rd quarter so 0.75 and so on. So it is a person-years contribution to one year. So, why the authors here decided to mid-year population of 2016 and the numerator covers 2 years? The mid-year population between 1st July 2016 and 30th June 2017 is something close to 1st January of 2017. Just a clarification please.

- State in the methods how the CID incidence confidence intervals were estimated.

- The statement starting as “The NHLS only test specimens for CD that conform to the shape of the container”. This is unclear and confusing. Please clarify.

- Figure 1 is important but it needs changes to 1) to separate clearly exclusions and 2) distinguish sample count from patient counts. Otherwise we will be confused on the calculations as one of the reviewers rightly asks. A sort of suggestion is attached as picture [pardon me for lack of technological sophistication, see the photo of a drawing attached].

- This paragraph related to figure 1 has a lot of information that belongs to results section. Please review.

- Be careful when interpreting odds-ratios as risk-ratios as in, for example, “receiving care in a specialized TB hospital resulted in a more than 11-fold increase in the risk of CDI”. This is incorrect. An example: the proportions of positivity among males and females are 19.12% and 18.28%, respectively. So the is [19.12/(100 – 19.12)]/[18.28/(100 – 18.28)] = 1.06 which is not a ratio of proportion 19.12/18.28 = 1.05. The difference between these two gets worse for higher OR.

III. Results

1. Table 1.

* The percentages cause confusion. I propose these changes

- Please add a row of totals

- For the 3 columns ‘n(%)’, ‘CDI’, and ‘non-CID’ make them the percentages be in columns

- Add one column named ‘CD positivity rate’ so the current percentage calculated in CDI would move to a column dedicated for row percentage.

- Why we need p-values here? Those p-values inform less than the ones on table 2.

* One of the reviewers complains about the numbers. Please re-verify the calculations.

- Eg: The positivity frequency for females 974/5,329 = 18.28% that is OK. But for males should be 756/3,953 = 19.12%.

* Why province is not here?

2. Table 2 –

* Why province is not here?

3. Table 3 needs similar changes as table 1.

4. Table 4 – please what is the unity of measurement of the incidence.

Remove the positivity rate. That is fine on table one.

Three columns are missing. One for the population and the numerators. That will be important in a future meta-analysis.

Add a row for the total. This is important to understand the 0.101 described in the results.

5. Table 5 – the same comments as for table 4.

be careful with abbreviation of CD.

6. The first paragraph of results ends with something as “On univariate analysis only level of patient care was statistically significantly associated with CDI (p < 0.001) (table 1)” Please do not use p-values as evidence of associations. For that, first describe the association measure.

7. If the p-value is below the 0.001 please write “p-value < 0.001” not “p = 0.000” as in “81% (OR 1.81 CI95% 1.51 – 2.17, p = 0.000)”.

8. All figures are numbered figure 1. Please correct this.

9. I am not sure about the seasonality based just with one year of data. Please note this limitation in your discussion

Journal Requirements:

2. Thank you for providing the date(s) when patient medical information was initially recorded. Please also include the date(s) on which your research team accessed the databases/records to obtain the retrospective data used in your study.

3. To comply with PLOS ONE submission guidelines, in your Methods section, please provide additional information regarding your statistical analyses. For more information on PLOS ONE's expectations for statistical reporting, please see https://journals.plos.org/plosone/s/submission-guidelines.#loc-statistical-reporting.

4. In your Methods section, please provide additional information about the participant recruitment method and the demographic details of your participants. Please ensure you have provided sufficient details to replicate the analyses such as:

a) a description in the methods section of any inclusion/exclusion criteria that were applied to participant selection,

b) a statement as to whether your sample can be considered representative of a larger population.

5. Thank you for stating in your ethics statement "The protocol for this study was reviewed by the Human Research Ethics Committee

(Medical) at the University of the Witwatersrand, Johannesburg and clearance was obtained (Ref: M200114) prior to commencing the study. This was a retrospective study utilizing secondary data sources. Prior to data analysis all unique identifiers were delinked to ensure anonymity." In your ethics statement, please clarify:

 - whether the ethics committee specifically approved this study.

 - whether consent was obtained

 - whether consent was informed

 - what type of consent you obtained (for instance, written or verbal, and if verbal, how it was documented and witnessed).

 - if your study included minors, state whether you obtained consent from parents or guardians.

 - if the need for consent was waived by the ethics committee, please include this information.

6. We note that you have indicated that data from this study are available upon request. PLOS only allows data to be available upon request if there are legal or ethical restrictions on sharing data publicly. For information on unacceptable data access restrictions, please see http://journals.plos.org/plosone/s/data-availability#loc-unacceptable-data-access-restrictions.

7. Please amend either the abstract on the online submission form (via Edit Submission) or the abstract in the manuscript so that they are identical.

8. We note that Figure 3 in your submission contain map images which may be copyrighted. All PLOS content is published under the Creative Commons Attribution License (CC BY 4.0), which means that the manuscript, images, and Supporting Information files will be freely available online, and any third party is permitted to access, download, copy, distribute, and use these materials in any way, even commercially, with proper attribution. For these reasons, we cannot publish previously copyrighted maps or satellite images created using proprietary data, such as Google software (Google Maps, Street View, and Earth). For more information, see our copyright guidelines: http://journals.plos.org/plosone/s/licenses-and-copyright.

8.1.    You may seek permission from the original copyright holder of Figure 1 to publish the content specifically under the CC BY 4.0 license. 

8.2.    If you are unable to obtain permission from the original copyright holder to publish these figures under the CC BY 4.0 license or if the copyright holder’s requirements are incompatible with the CC BY 4.0 license, please either i) remove the figure or ii) supply a replacement figure that complies with the CC BY 4.0 license. Please check copyright information on all replacement figures and update the figure caption with source information. If applicable, please specify in the figure caption text when a figure is similar but not identical to the original image and is therefore for illustrative purposes only.

9. Please include a caption for figures 2 and 3.

Reviewers' comments:

Reviewer's Responses to Questions

**Comments to the Author**

1. Is the manuscript technically sound, and do the data support the conclusions?

Reviewer #1: Yes

Reviewer #2: Yes

2. Has the statistical analysis been performed appropriately and rigorously? 

Reviewer #1: Yes

Reviewer #2: Yes

3. Have the authors made all data underlying the findings in their manuscript fully available?

Reviewer #1: Yes

Reviewer #2: Yes

4. Is the manuscript presented in an intelligible fashion and written in standard English?

Reviewer #1: Yes

Reviewer #2: Yes

5. Review Comments to the Author

Reviewer #1: This is a very interesting and informative article. This is the first study to determine the incidence of CDI in the South African public sector. All of data are well presented. Considering this, I suggest to accept this article

Reviewer #2: The authors of this paper have addressed key health concerns unique to African continent and a growing problem globally. The paper is well written, organized and easy to read. The English language is from a first speaking point and the paper reads well overall.

6. PLOS authors have the option to publish the peer review history of their article (what does this mean?). If published, this will include your full peer review and any attached files.

Reviewer #1: **Yes: **Fernando Gil

Reviewer #2: No

---

## [Author Response · Author response to Decision Letter 0]

23 Sep 2021

Dear Prof Augusto

Thank you for your positive response to our manuscript and, to the reviewers for their valuable suggestions to strengthen our paper. Thank for allowing us an extension on addressing the reviewer comments. We apologize for the delay in submitting a revised manuscript.

We have revised the manuscript based on the reviewers’ comments, including:

• Figure 1 has been extensively revised.

• The analysis of factors associated with Clostridioides difficile was redone. We agree that estimating incidence rate ratios rather than odds ratios utilizing logistic regression is more appropriate.

Attached you will find a detailed point-by-point response to each of the reviewers’ comments.

Best regards,

Pieter de Jager

2 August 2021

Fellow: Anaesthesiology, Mount Sinai Hospital, Toronto

MBChB(UFS), FCA(SA), MSc(LSE)

Email: ppdejager@gmail.com

---

## [Decision Letter · Decision Letter 1]

27 Oct 2021

Epidemiology of Clostridioides difficile in South Africa

PONE-D-21-03916R1

Dear Dr. de Jager,

We’re pleased to inform you that your manuscript has been judged scientifically suitable for publication and will be formally accepted for publication once it meets all outstanding technical requirements.

Kind regards,

Orvalho Augusto, MD, MPH

Academic Editor

PLOS ONE

Additional Editor Comments (optional):

The issues raised on the previous round have been address properly.

As previously noted, this is an important contribution for the body of literature on this not that rare infectious condition in times of a “pandemic” of antimicrobial resistance. I applaud the authors for this exercise.

Few more:

1. Please correct multivariate to multivariable in the whole document. Multivariate is usually reserved for situations simultaneous assessment of multiple outcomes, whereas multivariable is for situations of one outcome and multiple predictors as is the case here.

2. Good that the authors do present the association analysis through Poisson regressions (a tricky to approximate the log-binomial regression which is so prone to fail to converge). However, let’s be careful with the interpretation of exponentiated coefficients here. Because you do not use the SA population (as an offset) in the model, then you are not modelling the incidence rate in the same spirit of the incidence definition (as in the current table 4). These models are about the proportion of positivity so the coefficients are multiplicative changes of such proportions. Can you make a note on those lines, please? (or may use some other designation eg: ratio of positivity for table 2).

3. The current table 3 is built as in table 1. But the table 1, is supplemented with the current table 2 presenting an unadjusted and adjusted analysis. I would suggest to make an unadjusted and adjusted as well for this.

4. Line 203: please put citation for Stata 16.

5. The presentation of the results is curious. Please, start with the measures of frequency (the incidence) and its geographic distribution. Then move to the analysis of factors.

Reviewers' comments:

Reviewer's Responses to Questions

**Comments to the Author**

1. If the authors have adequately addressed your comments raised in a previous round of review and you feel that this manuscript is now acceptable for publication, you may indicate that here to bypass the “Comments to the Author” section, enter your conflict of interest statement in the “Confidential to Editor” section, and submit your "Accept" recommendation.

Reviewer #2: All comments have been addressed

2. Is the manuscript technically sound, and do the data support the conclusions?

Reviewer #2: Yes

3. Has the statistical analysis been performed appropriately and rigorously? 

Reviewer #2: Yes

4. Have the authors made all data underlying the findings in their manuscript fully available?

Reviewer #2: Yes

5. Is the manuscript presented in an intelligible fashion and written in standard English?

Reviewer #2: Yes

6. Review Comments to the Author

Reviewer #2: The previous review comments have been adequately addressed. The paper addresses key concerns about infection prevention and control mechanisms in health-care environment especially Clostridium difficile infections as a potential emerging global public health problem that needs urgent attention.

7. PLOS authors have the option to publish the peer review history of their article (what does this mean?). If published, this will include your full peer review and any attached files.

Reviewer #2: No

---

## [Editor Report · Acceptance letter]

15 Nov 2021

PONE-D-21-03916R1 

Epidemiology of *Clostridioides difficile* in South Africa 

Dear Dr. de Jager:

I'm pleased to inform you that your manuscript has been deemed suitable for publication in PLOS ONE. Congratulations! Your manuscript is now with our production department. 

Kind regards, 

on behalf of

Dr. Orvalho Augusto 

Academic Editor

PLOS ONE